# Minimax Estimation of Maximum Mean Discrepancy with Radial Kernels

**Ilya Tolstikhin**
Department of Empirical Inference
MPI for Intelligent Systems
Tübingen 72076, Germany
ilya@tuebingen.mpg.de

**Bharath K. Sriperumbudur**
Department of Statistics
Pennsylvania State University
University Park, PA 16802, USA
bks18@psu.edu

**Bernhard Schölkopf**
Department of Empirical Inference
MPI for Intelligent Systems
Tübingen 72076, Germany
bs@tuebingen.mpg.de

## Abstract

Maximum Mean Discrepancy (MMD) is a distance on the space of probability measures which has found numerous applications in machine learning and nonparametric testing. This distance is based on the notion of embedding probabilities in a reproducing kernel Hilbert space. In this paper, we present the first known lower bounds for the estimation of MMD based on finite samples. Our lower bounds hold for any radial universal kernel on $\mathbb{R}^d$ and match the existing upper bounds up to constants that depend only on the properties of the kernel. Using these lower bounds, we establish the minimax rate optimality of the empirical estimator and its $U$-statistic variant, which are usually employed in applications.

## 1 Introduction

Over the past decade, the notion of embedding probability measures in a Reproducing Kernel Hilbert Space (RKHS) [1, 13, 18, 17] has gained a lot of attention in machine learning, owing to its wide applicability. Some popular applications of RKHS embedding of probabilities include two-sample testing [5, 6], independence [7] and conditional independence testing [3], feature selection [14], covariate-shift [13], causal discovery [9], density estimation [15], kernel Bayes' rule [4], and distribution regression [20]. This notion of embedding probability measures can be seen as a generalization of classical kernel methods which deal with embedding points of an input space as elements in an RKHS. Formally, given a probability measure $P$ and a continuous positive definite real-valued kernel $k$ (we denote $\mathcal{H}$ to be the corresponding RKHS) defined on a separable topological space $\mathcal{X}$, $P$ is embedded into $\mathcal{H}$ as $\mu_P := \int k(\cdot, x)\, dP(x)$, called the *mean element* or the *kernel mean* assuming $k$ and $P$ satisfy $\int_{\mathcal{X}} \sqrt{k(x,x)}\, dP(x) < \infty$. Based on the above embedding of $P$, [5] defined a distance—called the *Maximum Mean Discrepancy* (MMD)—on the space of probability measures as the distance between the corresponding mean elements, i.e.,

$$\mathrm{MMD}_k(P, Q) = \|\mu_P - \mu_Q\|_{\mathcal{H}}.$$

We refer the reader to [18, 17] for a detailed study on the properties of MMD and its relation to other distances on probabilities.

**Estimation of kernel mean.** In all the above mentioned applications, since the only knowledge of the underlying distribution is through random samples drawn from it, an estimate of $\mu_P$ is employed

in practice. In applications such as two-sample test [5, 6] and independence test [7] that involve MMD, an estimate of MMD is constructed based on the estimates of $\mu_P$ and $\mu_Q$ respectively. The simple and most popular estimator of $\mu_P$ is the empirical estimator, $\mu_{P_n} := \frac{1}{n}\sum_{i=1}^n k(\cdot, X_i)$ which is a Monte Carlo approximation of $\mu_P$ based on random samples $(X_i)_{i=1}^n$ drawn i.i.d. from $P$. Recently, [10] proposed a shrinkage estimator of $\mu_P$ based on the idea of James-Stein shrinkage, which is demonstrated to empirically outperform $\mu_{P_n}$. While both these estimators are shown to be $\sqrt{n}$-consistent [13, 5, 10], it was not clear until the recent work of [21] whether any of these estimators are minimax rate optimal, i.e., is there an estimator of $\mu_P$ that yields a convergence rate faster than $n^{-1/2}$? Based on the minimax optimality of the sample mean (i.e., $\overline{X} := \frac{1}{n}\sum_{i=1}^n X_i$) for the estimation of a finite dimensional mean of a normal distribution at a minimax rate of $n^{-1/2}$ [8, Chapter 5, Example 1.14], while one can intuitively argue that the empirical and shrinkage estimators of $\mu_P$ are minimax rate optimal, it is difficult to extend the finite dimensional argument in a rigorous manner to the estimation of the infinite dimensional object, $\mu_P$. Note that $\mathcal{H}$ is infinite dimensional if $k$ is universal [19, Chapter 4], e.g., Gaussian kernel. By establishing a remarkable relation between the MMD of two Gaussian distributions and the Euclidean distance between their means for any bounded continuous translation invariant universal kernel on $\mathcal{X} = \mathbb{R}^d$, [21] rigorously showed that the estimation of $\mu_P$ is only as hard as the estimation of the finite dimensional mean of a normal distribution and thereby established the minimax rate of estimating $\mu_P$ to be $n^{-1/2}$. This in turn demonstrates the minimax rate optimality of empirical and shrinkage estimators of $\mu_P$.

**Estimation of MMD.** In this paper, we are interested in the minimax optimal estimation of $\mathrm{MMD}_k(P,Q)$. The question of finding optimal estimators of MMD is of interest in applications such as kernel-based two-sample [5] and independence tests [7] as the test statistic is indeed an estimate of MMD and it is important to use statistically optimal estimators in the construction of these kernel based tests. An estimator of MMD that is currently employed in these applications is based on the empirical estimators of $\mu_P$ and $\mu_Q$, i.e.,

$$\mathrm{MMD}_{n,m} := \|\mu_{P_n} - \mu_{Q_m}\|_{\mathcal{H}},$$

which is constructed from samples $(X_i)_{i=1}^n \overset{i.i.d.}{\sim} P$ and $(Y_i)_{i=1}^m \overset{i.i.d.}{\sim} Q$. [5, 7] also considered a $U$-statistic variant of $\mathrm{MMD}_{n,m}$ as a test statistic in these applications. As discussed above, while $\mu_{P_n}$ and $\mu_{Q_m}$ are minimax rate optimal estimators of $\mu_P$ and $\mu_Q$ respectively, they need not guarantee that $\mathrm{MMD}_{n,m}$ is minimax rate optimal. Using the fact that $\|\mu_{P_n} - \mu_P\|_{\mathcal{H}} = O_p(n^{-1/2})$ and

$$|\mathrm{MMD}_k(P,Q) - \mathrm{MMD}_{n,m}| \leq \|\mu_P - \mu_{P_n}\|_{\mathcal{H}} + \|\mu_{Q_m} - \mu_Q\|_{\mathcal{H}},$$

it is easy to see that

$$|\mathrm{MMD}_k(P,Q) - \mathrm{MMD}_{n,m}| = O_p(n^{-1/2} + m^{-1/2}). \tag{1}$$

In fact, if $k$ is a bounded kernel, it can be shown that the constants (which are hidden in the order notation in (1)) depend only on the bound on the kernel and are independent of $\mathcal{X}$, $P$ and $Q$. The goal of this work is to find the minimax rate $r_{n,m,k}(\mathcal{P})$ and a positive constant $c_k(\mathcal{P})$ (independent of $m$ and $n$) such that

$$\inf_{\hat{F}_{n,m}} \sup_{P,Q \in \mathcal{P}} P^n \times Q^m \left\{ r_{n,m,k}^{-1}(\mathcal{P}) \, |\hat{F}_{n,m} - \mathrm{MMD}_k(P,Q)| \geq c_k(\mathcal{P}) \right\} > 0, \tag{2}$$

where $\mathcal{P}$ is a suitable subset of Borel probability measures on $\mathcal{X}$, the infimum is taken over all estimators $\hat{F}_{n,m}$ mapping the i.i.d. sample $\{(X_i)_{i=1}^n, (Y_i)_{i=1}^m\}$ to $\mathbb{R}^+$, and $P^n \times Q^m$ denotes the probability measure associated with the sample when $(X_i)_{i=1}^n \overset{i.i.d.}{\sim} P$ and $(Y_i)_{i=1}^m \overset{i.i.d.}{\sim} Q$. In addition to the rate, we are also interested in the behavior of $c_k(\mathcal{P})$ in terms of its dependence on $k$, $\mathcal{X}$ and $\mathcal{P}$.

**Contributions.** The main contribution of the paper is in establishing $m^{-1/2} + n^{-1/2}$, i.e., $r_{n,m,k}(\mathcal{P}) = \sqrt{(m+n)/mn}$ as the minimax rate for estimating $\mathrm{MMD}_k(P,Q)$ when $k$ is a radial universal kernel (examples include the Gaussian, Matérn and inverse multiquadric kernels) on $\mathbb{R}^d$ and $\mathcal{P}$ is the set of all Borel probability measures on $\mathbb{R}^d$ with infinitely differentiable densities. This result guarantees that $\mathrm{MMD}_{n,m}$ and its $U$-statistic variant are minimax rate optimal estimators of $\mathrm{MMD}_k(P,Q)$, which thereby ensures the minimax optimality of the test statistics used in kernel two-sample and independence tests. We would like to highlight the fact that our result of the minimax lower bound on $\mathrm{MMD}_k(P,Q)$ implies part of the results of [21] related to the minimax estimation

of $\mu_P$, as it can be seen that any $\epsilon$-accurate estimators $\hat{\mu}_P$ and $\hat{\mu}_Q$ of $\mu_P$ and $\mu_Q$ respectively in the RKHS norm lead to the $2\epsilon$-accurate estimator $\hat{F}_{n,m} := \|\hat{\mu}_P - \hat{\mu}_Q\|_{\mathcal{H}}$ of $\mathrm{MMD}_k(P, Q)$, i.e.,

$$c_k(\mathcal{P})(n^{-1/2} + m^{-1/2}) \le |\mathrm{MMD}_k(P, Q) - \hat{F}_{n,m}| \le \|\mu_P - \hat{\mu}_P\|_{\mathcal{H}} + \|\mu_Q - \hat{\mu}_Q\|_{\mathcal{H}}.$$

In Section 2, we present the main results of our work wherein Theorem 1 is developed by employing the ideas of [21] involving Le Cam's method (see Theorem 3) [22, Sections 2.3 and 2.6]. However, we show that while the minimax rate is $m^{-1/2} + n^{-1/2}$, there is a sub-optimal dependence on $d$ in the constant $c_k(\mathcal{P})$ which makes the result uninteresting in high dimensional scenarios. To alleviate this issue, we present a refined result in Theorem 2 based on the method of two fuzzy hypotheses (see Theorem 4) [22, Section 2.7.4] which shows that $c_k(\mathcal{P})$ in (2) is independent of $d$ (i.e., $\mathcal{X}$). This result provides a sharp lower bound for MMD estimation both in terms of the rate and the constant (which is independent of $\mathcal{X}$) that matches with behavior of the upper bound for $\mathrm{MMD}_{n,m}$. The proofs of these results are provided in Section 3 while supplementary results are collected in an appendix.

**Notation.** In this work we focus on *radial kernels*, i.e., $k(x, y) = \psi(\|x - y\|^2)$ for all $x, y \in \mathbb{R}^d$. Schoenberg's theorem [12] states that a radial kernel $k$ is positive definite for every $d$ if and only if there exists a non-negative finite Borel measure $\nu$ on $[0, \infty)$ such that

$$k(x, y) = \int_0^\infty e^{-t\|x-y\|^2} \, d\nu(t) \tag{3}$$

for all $x, y \in \mathbb{R}^d$. An important example of a radial kernel is the *Gaussian kernel* $k(x, y) = \exp\{-\|x - y\|^2/(2\eta^2)\}$ for $\eta^2 > 0$. [17, Proposition 5] showed that $k$ in (3) is universal if and only if $\mathrm{supp}(\nu) \ne \{0\}$, where for a finite non-negative Borel measure $\mu$ on $\mathbb{R}^d$ we define $\mathrm{supp}(\mu) = \{x \in \mathbb{R}^d \,|\, \text{if } x \in U \text{ and } U \text{ is open then } \mu(U) > 0\}$.

## 2 Main results

In this section, we present the main results of our work wherein we develop minimax lower bounds for the estimation of $\mathrm{MMD}_k(P, Q)$ when $k$ is a radial universal kernel on $\mathbb{R}^d$. We show that the minimax rate for estimating $\mathrm{MMD}_k(P, Q)$ based on random samples $(X_i)_{i=1}^n \overset{i.i.d.}{\sim} P$ and $(Y_i)_{i=1}^m \overset{i.i.d.}{\sim} Q$ is $m^{-1/2} + n^{-1/2}$, thereby establishing the minimax rate optimality of the empirical estimator $\mathrm{MMD}_{n,m}$ of $\mathrm{MMD}(P, Q)$. First, we present the following result (proved in Section 3.1) for Gaussian kernels, which is based on an argument similar to the one used in [21] to obtain a minimax lower bound for the estimation of $\mu_P$.

**Theorem 1.** *Let $\mathcal{P}$ be the set of all Borel probability measures over $\mathbb{R}^d$ with continuously infinitely differentiable densities. Let $k$ be a Gaussian kernel with bandwidth parameter $\eta^2 > 0$. Then the following holds:*

$$\inf_{\hat{F}_{n,m}} \sup_{P,Q \in \mathcal{P}} P^n \times Q^m \left\{ \left| \mathrm{MMD}_k(P, Q) - \hat{F}_{n,m} \right| \ge \frac{1}{8} \sqrt{\frac{1}{d+1}} \max \left\{ \frac{1}{\sqrt{n}}, \frac{1}{\sqrt{m}} \right\} \right\} \ge \frac{1}{5}. \tag{4}$$

The following remarks can be made about Theorem 1.

*(a)* Theorem 1 shows that $\mathrm{MMD}_k(P, Q)$ cannot be estimated at a rate faster than $\max\{n^{-1/2}, m^{-1/2}\}$ by any estimator $\hat{F}_{n,m}$ for all $P, Q \in \mathcal{P}$. Since $\max\{m^{-1/2}, n^{-1/2}\} \ge \frac{1}{2}(m^{-1/2} + n^{-1/2})$, the result combined with (1) therefore establishes the minimax rate optimality of the empirical estimator, $\mathrm{MMD}_{n,m}$.

*(b)* While Theorem 1 shows the right order of dependence on $m$ and $n$, the dependence on $d$ seems to be sub-optimal as the upper bound on $|\mathrm{MMD}_{n,m} - \mathrm{MMD}_k(P, Q)|$ depends only on the bound on the kernel and is independent of $d$. This sub-optimal dependence on $d$ may be due to the fact the proof of Theorem 1 (see Section 3.1) as aforementioned is closely based on the arguments applied in [21] for the minimax estimation of $\mu_P$. While the lower bounding technique used in [21]—which is commonly known as Le Cam's method based on *many hypotheses* [22, Chapter 2]—provides optimal results in the problem of estimation of *functions* (e.g., estimation of $\mu_P$ in the norm of $\mathcal{H}$), it often fails to do so in the case of estimation of real-valued *functionals*, which is precisely the focus of our work. Even though Theorem 1 is sub-optimal, we presented the result to highlight the fact that

the minimax lower bounds for estimation of $\mu_P$ may not yield optimal results for $\mathrm{MMD}_k(P,Q)$. In Theorem 2, we will develop a new argument based on *two fuzzy hypotheses*, which is a method of choice for nonparametric estimation of *functionals* [22, Section 2.7.4]. This will allow us to get rid of the superfluous dependence on the dimensionality $d$ in the lower bound.

*(c)* While Theorem 1 holds for only Gaussian kernels, we would like to mention that by using the analysis of [21], Theorem 1 can be straightforwardly improved in various ways: (i) it can be generalized to hold for a wide class of *radial universal kernels*, (ii) the factor $d^{-1/2}$ in (4) can be removed altogether for the case when $\mathcal{P}$ consists of all Borel *discrete* distributions on $\mathbb{R}^d$. However, these improvements do not involve any novel ideas than those captured by the proof of Theorem 1 and so will not be discussed in this work. For details, we refer an interested reader to Theorems 2 and 6 of [21] for extension to radial universal kernels and discrete measures, respectively.

*(d)* Finally, it is worth mentioning that any lower bound on the minimax probability (including the bounds of Theorems 1 and 2) leads to a lower bound on the *minimax risk*, which is based on a simple application of the Markov's inequality: $\mathbb{E}_{P^n \times Q^m} \left[ s_{n,m}^{-1} \cdot |A_{n,m}| \right] \geq P^n \times Q^m \{|A_{n,m}| \geq s_{n,m}\}$.

The following result (proved in Section 3.2) is the main contribution of this work. It provides a minimax lower bound for the problem of MMD estimation, which holds for general radial universal kernels. In contrast to Theorem 1, it avoids the superfluous dependence on $d$ and depends only on the properties of $k$ while exhibiting the correct rate.

**Theorem 2.** *Let $\mathcal{P}$ be the set of all Borel probability measures over $\mathbb{R}^d$ with continuously infinitely differentiable densities. Let $k$ be a radial kernel on $\mathbb{R}^d$ of the form* (3)*, where $\nu$ is a bounded non-negative measure on $[0,\infty)$. Assume that there exist $0 < t_0 \leq t_1 < \infty$ and $0 < \beta < \infty$ such that $\nu([t_0, t_1]) \geq \beta$. Then the following holds:*

$$\inf_{\hat{F}_{n,m}} \sup_{P,Q \in \mathcal{P}} P^n \times Q^m \left\{ \left| \mathrm{MMD}_k(P,Q) - \hat{F}_{n,m} \right| \geq \frac{1}{20} \sqrt{\frac{\beta t_0}{t_1 e}} \max \left\{ \frac{1}{\sqrt{n}}, \frac{1}{\sqrt{m}} \right\} \right\} \geq \frac{1}{14}. \quad (5)$$

Note that the existence of $0 < t_0 \leq t_1 < \infty$ and $0 < \beta < \infty$ such that $\nu([t_0, t_1]) \geq \beta$ ensures that $\mathrm{supp}(\nu) \neq \{0\}$ (i.e., the kernel is not a constant function), which implies $k$ is universal. If $k$ is a Gaussian kernel with bandwidth parameter $\eta^2 > 0$, it is easy to verify that $t_0 = t_1 = (2\eta^2)^{-1}$ and $\beta = 1$ satisfy $\nu([t_0, t_1]) \geq \beta$ as the Gaussian kernel is generated by $\nu = \delta_{1/(2\eta^2)}$ in (3), where $\delta_x$ is a Dirac measure supported at $x$. Therefore we obtain a dimension independent constant in (5) for Gaussian kernels compared to the bound in (4).

## 3  Proofs

In this section, we present the proofs of Theorems 1 and 2. Before we present the proofs, we first introduce the setting of nonparametric estimation. Let $F \colon \Theta \to \mathbb{R}$ be a *functional* defined on a measurable space $\Theta$ and $\mathcal{P}_\Theta = \{P_\theta \colon \theta \in \Theta\}$ be a family of probability distributions indexed by $\Theta$ and defined over a measurable space $\mathcal{X}$ associated with data. We observe the data $D \in \mathcal{X}$ distributed according to an unknown element $P_\theta \in \mathcal{P}_\Theta$ and the goal is to estimate $F(\theta)$. Usually $\mathcal{X}$, $D$, and $P_\theta$ will depend on sample size $n$. Let $\hat{F}_n := \hat{F}_n(D)$ be an estimator of $F(\theta)$ based on $D$. The following well known result [22, Theorem 2.2] provides a lower bound on the minimax probability of this problem. We refer the reader to Appendix A for a proof of its more general version.

**Theorem 3.** *Assume there exist $\theta_0, \theta_1 \in \Theta$ such that $|F(\theta_0) - F(\theta_1)| \geq 2s > 0$ and $\mathrm{KL}(P_{\theta_1} \| P_{\theta_0}) \leq \alpha$ with $0 < \alpha < \infty$. Then*

$$\inf_{\hat{F}_n} \sup_{\theta \in \Theta} P_\theta \left\{ |\hat{F}_n(D) - F(\theta)| \geq s \right\} \geq \max \left( \frac{1}{4} e^{-\alpha}, \frac{1 - \sqrt{\alpha/2}}{2} \right),$$

*where $\mathrm{KL}(P_{\theta_1} \| P_{\theta_0}) := \int \log \left( \frac{dP_{\theta_1}}{dP_{\theta_0}} \right) dP_{\theta_1}$ denotes the Kullback-Leibler divergence between $P_{\theta_1}$ and $P_{\theta_0}$.*

The above result (also called the *Le Cam's method*) provides the recipe for obtaining minimax lower bounds, where the goal is to construct two *hypotheses* $\theta_0, \theta_1 \in \Theta$ such that (i) $F(\theta_0)$ and $F(\theta_1)$ are far apart, while (ii) the corresponding distributions, $P_{\theta_0}$ and $P_{\theta_1}$ are close enough. The requirement (i) can be relaxed by introducing two random (*fuzzy*) hypotheses $\theta_0, \theta_1 \in \Theta$, and requiring $F(\theta_0)$

and $F(\theta_1)$ to be far apart *with high probability*. This weaker requirement leads to a lower bounding technique, called the *method of two fuzzy hypotheses*. This method is captured by the following theorem [22, Theorem 2.14] and is commonly used to derive lower bounds on the minimax risk in the problem of estimation of functionals [22, Section 2.7.4].

**Theorem 4.** *Let $\mu_0$ and $\mu_1$ be any probability distributions over $\Theta$. Assume that*

1. *There exist $c \in \mathbb{R}$, $s > 0$, $0 \le \beta_0, \beta_1 < 1$ such that $\mu_0\big(\theta \colon F(\theta) \le c\big) \ge 1 - \beta_0$ and $\mu_1\big(\theta \colon F(\theta) \ge c + 2s\big) \ge 1 - \beta_1$.*

2. *There exist $\tau > 0$ and $0 < \alpha < 1$ such that $\mathbb{P}_1\left(\frac{d\mathbb{P}_0^a}{d\mathbb{P}_1} \ge \tau\right) \ge 1 - \alpha$, where*

$$\mathbb{P}_i(D) = \int P_\theta(D)\mu_i(d\theta), \quad i \in \{0, 1\}$$

*and $\mathbb{P}_0^a$ is the absolutely continuous component of $\mathbb{P}_0$ with respect to $\mathbb{P}_1$.*

*Then*

$$\inf_{\hat{F}_n} \sup_{\theta \in \Theta} P_\theta \left\{ |\hat{F}_n(D) - F(\theta)| \ge s \right\} \ge \frac{\tau(1 - \alpha - \beta_1) - \beta_0}{1 + \tau}.$$

With this set up and background, we are ready to prove Theorems 1 and 2.

### 3.1 Proof of Theorem 1

The proof is based on Theorem 3 and treats two cases $m \ge n$ and $m < n$ separately. We consider only the case $m \ge n$ as the second one follows the same steps. Let $\mathcal{G}_d$ denote a class of multivariate Gaussian distributions over $\mathbb{R}^d$ with covariance matrices proportional to identity matrix $I_d \in \mathbb{R}^{d \times d}$. In our case $\mathcal{G}_d \subseteq \mathcal{P}$, which leads to the following lower bound for any $s > 0$:

$$\sup_{P,Q \in \mathcal{P}} P^n \times Q^m \left\{ \left| \mathrm{MMD}_k(P, Q) - \hat{F}_{n,m} \right| \ge s \right\} \ge \sup_{P,Q \in \mathcal{G}_d} P^n \times Q^m \left\{ \left| \mathrm{MMD}_k(P, Q) - \hat{F}_{n,m} \right| \ge s \right\}.$$

Note that every element $G(\mu, \sigma^2 I_d) \in \mathcal{G}_d$ is indexed by a pair $(\mu, \sigma^2) \in \mathbb{R}^d \times (0, \infty) =: \tilde{\Theta}$. Given two elements $P, Q \in \mathcal{G}_d$, the data is distributed according to $P^n \times Q^m$. This brings us into the context of Theorem 3 with $\Theta := \tilde{\Theta} \times \tilde{\Theta}$, $\mathcal{X} := (\mathbb{R}^d)^{n+m}$, $P_\theta := G_1^n \times G_2^m$ for $\theta = (\tilde{\theta}_1, \tilde{\theta}_2) \in \Theta$ with Gaussian distributions $G_1$ and $G_2$ corresponding to parameters $\tilde{\theta}_1, \tilde{\theta}_2 \in \tilde{\Theta}$ respectively, and $F(\theta) = \mathrm{MMD}_k(G_1, G_2)$.

In order to apply Theorem 3 we need to choose two probability distributions $P_{\theta_0}$ and $P_{\theta_1}$. We define four different $d$-dimensional Gaussian distributions:

$$P_0 = G(\mu_0^P, \sigma^2 I_d), \quad Q_0 = G(\mu_0^Q, \sigma^2 I_d), \quad P_1 = Q_1 = G(0, \sigma^2 I_d)$$

with

$$\sigma^2 = \frac{c_1 \eta^2}{d}\left(2 + \frac{n}{m}\right), \quad \|\mu_0^P\|^2 = \frac{c_2 \eta^2}{d}\left(\frac{1}{n} + \frac{1}{m}\right), \quad \|\mu_0^Q\|^2 = \frac{c_2 \eta^2}{dm}, \quad \|\mu_0^P - \mu_0^Q\|^2 = \frac{c_3 \eta^2}{dn},$$

where $c_1, c_2, c_3 > 0$ are positive constants independent of $m$ and $n$ to be specified later. Note that this construction is possible as long as $\sqrt{\frac{c_3}{n}} \le \sqrt{c_2\left(\frac{1}{n} + \frac{1}{m}\right)} + \sqrt{\frac{c_2}{m}}$, which is clearly satisfied if $c_3 \le c_2$.

First we will check the upper bound on the KL divergence between the distributions. Using the chain rule of KL divergence and its closed form expression for Gaussian distributions we write

$$\mathrm{KL}(P_1^n \times Q_1^m \| P_0^n \times Q_0^m) = n \cdot \frac{\|\mu_0^P\|^2}{2\sigma^2} + m \cdot \frac{\|\mu_0^Q\|^2}{2\sigma^2} = n \cdot \frac{c_2 \eta^2 \left(\frac{1}{n} + \frac{1}{m}\right)}{2c_1 \eta^2 \left(2 + \frac{n}{m}\right)} + m \cdot \frac{c_2 \eta^2 \frac{1}{m}}{2c_1 \eta^2 \left(2 + \frac{n}{m}\right)}$$

$$= c_2 \frac{2 + \frac{n}{m}}{2c_1 \left(2 + \frac{n}{m}\right)} = \frac{c_2}{2c_1}.$$

Next we need to lower bound an absolute value between $\mathrm{MMD}_k(P_0, Q_0)$ and $\mathrm{MMD}_k(P_1, Q_1)$. Note that

$$|\mathrm{MMD}_k(P_0, Q_0) - \mathrm{MMD}_k(P_1, Q_1)| = \mathrm{MMD}_k(P_0, Q_0). \tag{6}$$

Using a closed-form expression for the MMD between Gaussian distribution [21, Eq. 25] we write

$$\mathrm{MMD}_k^2(P_0, Q_0) = 2\left(\frac{\eta^2}{\eta^2 + 2\sigma^2}\right)^{d/2}\left(1 - \exp\left(-\frac{\|\mu_0^P - \mu_0^Q\|^2}{2\eta^2 + 4\sigma^2}\right)\right).$$

Assume

$$\frac{\|\mu_0^P - \mu_0^Q\|^2}{2\eta^2 + 4\sigma^2} \le 1. \tag{7}$$

Using $1 - e^{-x} \ge x/2$, which holds for $x \in [0, 1]$, we write

$$|\mathrm{MMD}_k(P_0, Q_0) - \mathrm{MMD}_k(P_1, Q_1)| \ge \left(\frac{d}{d + 2c_1\left(2 + \frac{n}{m}\right)}\right)^{d/4}\sqrt{\frac{\|\mu_0^P - \mu_0^Q\|^2}{2\eta^2 + 4\sigma^2}}.$$

Since $m \ge n$ and $(1 - \frac{1}{x})^{x-1}$ monotonically decreases to $e^{-1}$ for $x \ge 1$, we have

$$\left(\frac{d}{d + 2c_1\left(2 + \frac{n}{m}\right)}\right)^{\frac{d}{4}} \ge \left(\frac{d}{d + 6c_1}\right)^{\frac{d}{4}} = \left(\left(1 - \frac{1}{1 + d/(6c_1)}\right)^{(1 + d/(6c_1) - 1)}\right)^{\frac{6c_1}{d}\cdot\frac{d}{4}} \ge e^{-\frac{3c_1}{2}}.$$

Using this and setting $c_3 = c_2$ we get

$$|\mathrm{MMD}_k(P_0, Q_0) - \mathrm{MMD}_k(P_1, Q_1)| \ge \frac{1}{\sqrt{n}}e^{-\frac{3c_1}{2}}\sqrt{\frac{c_2}{2d + 4c_1\left(2 + \frac{n}{m}\right)}} \ge \frac{1}{\sqrt{n}}e^{-\frac{3c_1}{2}}\sqrt{\frac{c_2}{2d + 12c_1}}.$$

Now we set $c_1 = 0.16$, $c_2 = 0.23$. Checking that Condition (7) is satisfied and noting that

$$\max\left(\frac{1}{4}e^{-\frac{c_2}{2c_1}}, \frac{1 - \sqrt{c_2/(4c_1)}}{2}\right) > \frac{1}{5}, \quad \frac{1}{2}e^{-\frac{3c_1}{2}}\sqrt{\frac{c_2}{2}} > \frac{1}{8} \quad \text{and} \quad \frac{1}{d + 6c_1} > \frac{1}{d + 1}$$

we conclude the proof with an application of Theorem 3.

## 3.2 Proof of Theorem 2

First, we repeat the argument presented in the proof of Theorem 1 to bring ourselves into the context of minimax estimation, introduced in the beginning of Section 3.1. Namely, we reduce the class of distributions $\mathcal{P}$ to its subset $\mathcal{G}_d$ containing all the multivariate Gaussian distributions over $\mathbb{R}^d$ with covariance matrices proportional to identity matrix $I_d \in \mathbb{R}^{d \times d}$. The proof is based on Theorem 4 and treats two cases $m \ge n$ and $m < n$ separately. We consider only the case $m \ge n$ as the second one follows the same steps.

In order to apply Theorem 4 we need to choose two "fuzzy hypotheses", that is two probability distributions $\mu_0$ and $\mu_1$ over $\Theta$. In our setting there is a one-to-one correspondence between parameters $\theta \in \Theta$ and pairs of Gaussian distributions $(G_1, G_2) \in \mathcal{G}_d \times \mathcal{G}_d$. Throughout the proof it will be more convenient to treat $\mu_0$ and $\mu_1$ as distributions over $\mathcal{G}_d \times \mathcal{G}_d$. We will set $\mu_0$ to be a Dirac measure supported on $(P_0, Q_0)$ with $P_0 = Q_0 = G(0, \sigma^2 I_d)$. Clearly, $\mathrm{MMD}_k(P_0, Q_0) = 0$. This gives

$$\mu_0\big(\theta \colon F(\theta) = 0\big) = 1$$

and the first inequality of Condition 1 in Theorem 4 holds with $c = 0$ and $\beta_0 = 0$. Next we set $\mu_1$ to be a distribution of a random pair $(P, Q)$ with

$$Q = G_d(0, \sigma^2 I_d), \quad P = G_d(\mu, \sigma^2 I_d), \quad \sigma^2 = \frac{1}{2t_1 d},$$

where $\mu \sim P_\mu$ for some probability distribution $P_\mu$ over $\mathbb{R}^d$ to be specified later. Next we are going to check Condition 2 of Theorem 4. For $D = (x_1, \ldots, x_n, y_1, \ldots, y_m)$ define "posterior" distributions

$$\mathbb{P}_i(D) = \int P_\theta(D)\mu_i(d\theta), \quad i \in \{0, 1\}$$

as in Theorem 4. Using Markov's inequality we write

$$\mathbb{P}_1\left(\frac{d\mathbb{P}_0}{d\mathbb{P}_1} < \tau\right) = \mathbb{P}_1\left(\frac{d\mathbb{P}_1}{d\mathbb{P}_0} > \tau^{-1}\right) \le \tau\mathbb{E}_1\left[\frac{d\mathbb{P}_1}{d\mathbb{P}_0}\right]. \tag{8}$$

We have

$$\frac{d\mathbb{P}_1}{d\mathbb{P}_0}(D) = \frac{\int_{\mathbb{R}^d} \prod_{j=1}^n e^{-\frac{\|x_j-\mu\|^2}{2\sigma^2}} \prod_{k=1}^m e^{-\frac{\|y_k\|^2}{2\sigma^2}} dP_\mu(\mu)}{\prod_{j=1}^n e^{-\frac{\|x_j\|^2}{2\sigma^2}} \prod_{k=1}^m e^{-\frac{\|y_k\|^2}{2\sigma^2}}} = \int_{\mathbb{R}^d} e^{-\frac{n\|\mu\|^2}{2\sigma^2}} e^{\frac{\langle \sum_{j=1}^n x_j, \mu \rangle}{\sigma^2}} dP_\mu(\mu).$$

Now we compute the expected value appearing in (8):

$$\mathbb{E}_{D\sim\mathbb{P}_1}\left[\frac{d\mathbb{P}_1}{d\mathbb{P}_0}(D)\right] = \int_{\mathbb{R}^d} e^{-\frac{n\|\mu\|^2}{2\sigma^2}} \mathbb{E}_{D\sim\mathbb{P}_1}\left[e^{\langle \sum_{j=1}^n x_j, \mu \rangle / \sigma^2}\right] dP_\mu(\mu)$$

$$= \int_{\mathbb{R}^d} e^{-\frac{n\|\mu\|^2}{2\sigma^2}} \left(\int_{\mathbb{R}^d} \mathbb{E}\left[e^{\frac{1}{\sigma^2}\langle \sum_{j=1}^n X_j^{\mu'}, \mu \rangle}\right] dP_\mu(\mu')\right) dP_\mu(\mu), \qquad (9)$$

where $X_1^{\mu'}, \ldots, X_n^{\mu'}$ are independent and distributed according to $G_d(\mu', \sigma^2 I_d)$. Note that $\sum_{j=1}^n X_j^{\mu'} \sim G_d(n\mu', n\sigma^2 I_d)$ and as a result $\left\langle \sum_{j=1}^n X_j^{\mu'}, \mu \right\rangle \sim G(n\langle\mu',\mu\rangle, n\sigma^2\|\mu\|^2)$. Using the closed form for the moment generating function of a Gaussian distribution $Z \sim G(\mu, \sigma^2)$, $\mathbb{E}\left[e^{tZ}\right] = e^{\mu t} e^{\frac{1}{2}\sigma^2 t^2}$, we get

$$\mathbb{E}\left[e^{\frac{1}{\sigma^2}\langle \sum_{j=1}^n X_j^{\mu'}, \mu \rangle}\right] = e^{\frac{n\langle\mu',\mu\rangle}{\sigma^2}} e^{\frac{n\|\mu\|^2}{2\sigma^2}}.$$

Together with (9) this gives

$$\mathbb{E}_{D\sim\mathbb{P}_1}\left[\frac{d\mathbb{P}_1}{d\mathbb{P}_0}(D)\right] = \int_{\mathbb{R}^d} e^{-\frac{n\|\mu\|^2}{2\sigma^2}} \left(\int_{\mathbb{R}^d} e^{\frac{n\langle\mu',\mu\rangle}{\sigma^2}} e^{\frac{n\|\mu\|^2}{2\sigma^2}} dP_\mu(\mu')\right) dP_\mu(\mu) = \mathbb{E}\left[e^{\frac{n\langle\mu',\mu\rangle}{\sigma^2}}\right], \quad (10)$$

where $\mu$ and $\mu'$ are independent random variables both distributed according to $P_\mu$. Now we set $P_\mu$ to be a uniform distribution in the $d$-dimensional cube of appropriate size

$$P_\mu := U\left[-c_1/\sqrt{dnt_1}, c_1/\sqrt{dnt_1}\right]^d.$$

In this case, using Lemma B.1 presented in Appendix B we get

$$\mathbb{E}\left[e^{\frac{n\langle\mu',\mu\rangle}{\sigma^2}}\right] = \prod_{i=1}^d \mathbb{E}\left[e^{\frac{n\mu_i\mu_i'}{\sigma^2}}\right] = \prod_{i=1}^d \frac{dn\sigma^2 t_1}{2nc_1^2} \mathrm{Shi}\left(\frac{n}{\sigma^2}\frac{c_1^2}{dnt_1}\right) = \left(\frac{1}{4c_1^2}\mathrm{Shi}\left(2c_1^2\right)\right)^d.$$

Using (10) and also assuming

$$\frac{1}{4c_1^2}\mathrm{Shi}\left(2c_1^2\right) \leq 1 \qquad (11)$$

we get

$$\mathbb{E}_{D\sim\mathbb{P}_1}\left[\frac{d\mathbb{P}_1}{d\mathbb{P}_0}(D)\right] \leq \frac{1}{4c_1^2}\mathrm{Shi}\left(2c_1^2\right).$$

Combining with (8) we finally get $\mathbb{P}_1\left(\frac{d\mathbb{P}_0}{d\mathbb{P}_1} < \tau\right) \leq \frac{\tau}{4c_1^2}\mathrm{Shi}\left(2c_1^2\right)$ or equivalently $\mathbb{P}_1\left(\frac{d\mathbb{P}_0}{d\mathbb{P}_1} \geq \tau\right) \geq 1 - \frac{\tau}{4c_1^2}\mathrm{Shi}\left(2c_1^2\right)$. This shows that Condition 2 of Theorem 4 is satisfied with $\alpha = \frac{\tau}{4c_1^2}\mathrm{Shi}\left(2c_1^2\right)$.

Finally, we need to check the second inequality of Condition 1 in Theorem 4. Take two Gaussian distributions $P = G_d(\mu, \sigma^2 I_d)$ and $Q = G_d(0, \sigma^2 I_d)$. Using [21, Eq. 30] we have

$$\mathrm{MMD}_k^2(P,Q) \geq \frac{\beta t_0}{e}\left(1 - \frac{2}{2+d}\right)\|\mu\|^2$$

given

$$\sigma^2 = \frac{1}{2t_1 d} \quad \text{and} \quad t_1\|\mu\|^2 \leq 1 + 4t_1\sigma^2. \qquad (12)$$

Notice that the largest diagonal of a $d$-dimensional cube scales as $\sqrt{d}$. Using this we conclude that for $\mu \sim P_\mu$ with probability 1 it holds that $\|\mu\|^2 \leq \frac{c_1^2}{t_1 n}$ and the second condition in (12) holds as long as $c_1^2 \leq n$. Using this we get for any $c_2 > 0$

$$\mathop{\mathbb{P}}_{(P,Q)\sim\mu_1}\left\{\mathrm{MMD}_k(P,Q) \geq c_2\sqrt{\frac{\beta t_0}{t_1 en}}\right\} \geq \mathop{\mathbb{P}}_{\mu\sim P_\mu}\left\{\|\mu\|^2 \geq \frac{c_2^2}{t_1 n}\left(\frac{2+d}{d}\right)\right\}. \qquad (13)$$

Note that for $\mu \sim P_\mu$, $\|\mu\|^2 = \sum_{i=1}^d \mu_i^2$ is a sum of $d$ i.i.d. bounded random variables. Also simple computations show that

$$\mathbb{E}\|\mu\|^2 = \sum_{i=1}^d \mathbb{E}\mu_i^2 = d\frac{c_1^2}{3dnt_1} = \frac{c_1^2}{3nt_1} \qquad \text{and} \qquad \mathbb{V}\|\mu\|^2 = \sum_{i=1}^d \mathbb{V}\mu_i^2 = \frac{4c_1^4}{45dn^2t_1^2}.$$

Using Chebyshev-Cantelli's inequality of Theorem B.2 (Appendix B) we get for any $\epsilon > 0$

$$\mathbb{P}_{\mu \sim P_\mu}\left\{\|\mu\|^2 \geq \mathbb{E}\|\mu\|^2 - \epsilon\right\} = 1 - \mathbb{P}_{\mu \sim P_\mu}\left\{-\|\mu\|^2 > -\mathbb{E}\|\mu\|^2 + \epsilon\right\} \geq 1 - \frac{1}{1 + \frac{45dn^2t_1^2}{4c_1^4}\epsilon^2}$$

or equivalently for any $\epsilon > 0$,

$$\mathbb{P}_{\mu \sim P_\mu}\left\{\|\mu\|^2 \geq c_1^2\left(\frac{1}{3} - \frac{2\epsilon}{3\sqrt{5d}}\right)\frac{1}{nt_1}\right\} \geq 1 - \frac{1}{1 + \epsilon^2}.$$

Choosing $\epsilon \leq \frac{\sqrt{5}}{2} - \frac{9\sqrt{5}}{2}\left(\frac{c_2}{c_1}\right)^2$, we can further lower bound (13):

$$\mathbb{P}_{(P,Q) \sim \mu_1}\left\{\text{MMD}_k(P,Q) \geq c_2\sqrt{\frac{\beta t_0}{t_1 en}}\right\} \geq \mathbb{P}_{\mu \sim P_\mu}\left\{\|\mu\|^2 \geq c_1^2\left(\frac{1}{3} - \frac{2\epsilon}{3\sqrt{5d}}\right)\frac{1}{nt_1}\right\} \geq 1 - \frac{1}{1 + \epsilon^2}.$$

We finally set $\tau = 0.4$, $c_1 = 0.8$, $c_2 = 0.1$, $\epsilon = \frac{\sqrt{5}}{2} - \frac{9\sqrt{5}}{2}\left(\frac{c_2}{c_1}\right)^2$, and check that inequality (11) and the second condition of (12) are satisfied, while

$$\frac{\tau\left(1 - \frac{\tau}{4c_1^2}\text{Shi}\left(2c_1^2\right) - \frac{1}{1+\epsilon^2}\right)}{1 + \tau} > \frac{1}{14}.$$

We complete the proof by application of Theorem 4.

## 4 Discussion

In this paper, we provided the first known lower bounds for the estimation of maximum mean discrepancy (MMD) based on finite random samples. Based on this result, we established the minimax rate optimality of the empirical estimator. Interestingly, we showed that for radial kernels on $\mathbb{R}^d$, the optimal speed of convergence depends only on the properties of the kernel and is independent of $d$. However, the paper does not address an important question about the minimax rates for MMD based tests. We believe that the minimax rates of testing with MMD matches with that of the minimax rates for MMD estimation and we intend to build on this work in future to establish minimax testing results involving MMD.

Since MMD is an integral probability metric (IPM) [11], a related problem of interest is the minimax estimation of IPMs. IPM is a class of distances on probability measures, which is defined as $\gamma(P,Q) := \sup\{\int f(x)\,d(P-Q)(x) : f \in \mathcal{F}\}$, where $\mathcal{F}$ is a class of bounded measurable functions on a topological space $\mathcal{X}$ with $P$ and $Q$ being Borel probability measures. It is well known [16] that the choice of $\mathcal{F} = \{f \in \mathcal{H} : \|f\|_{\mathcal{H}} \leq 1\}$ yields $\text{MMD}_k(P,Q)$ where $\mathcal{H}$ is a reproducing kernel Hilbert space with a bounded reproducing kernel $k$. [16] studied the empirical estimation of $\gamma(P,Q)$ for various choices of $\mathcal{F}$ and established the consistency and convergence rates for the empirical estimator. However, it remains an open question as to whether these rates are minimax optimal.

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
