[Supplementary Material · nips16mmdminimax_supp.pdf]

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

## A  Minimax lower bounds: Method based on many hypotheses

In Section 3 we already introduced the setting of a nonparametric estimation of a functional and presented a lower bounding method involving two hypotheses (see Theorem 3). In this appendix, we present a more general result along with a sketch of its proof for obtaining minimax lower bounds using *many hypotheses* (i.e., more than two).

The setting introduced in Section 3 is slightly different from the one introduced in Section 2.1 of [22] and studied throughout the book. In short, a standard estimation problem considered in nonparametric statistics deals with estimation of a *function* $\theta$ (for instance, a probability density function underlying $P_\theta$) in certain metric spaces, such as $L_2([0,1])$ or $L_2(\mathbb{R}^d)$. Meanwhile, our work deals with estimation of a real valued parameter $F(\theta) \in \mathbb{R}$ (i.e., MMD between two distributions), or in other words an estimation of a *functional*. Results summarized in this section are presented in [22] in the former context of *function estimation*. For this reason we present them with a short sketch of proof. The following result is a particular instance of [22, Theorem 2.5].

**Theorem A.1** (Lower bound based on many hypotheses). *Assume $M \geq 2$ and suppose that there exist $\theta_0, \ldots, \theta_M \in \Theta$ such that (i) $|F(\theta_i) - F(\theta_j)| \geq 2s > 0$ for all $0 \leq i < j \leq M$; (ii) $P_{\theta_i}$ is absolutely continuous w.r.t. $P_{\theta_0}$ for all $i = 1, \ldots, M$, and $\frac{1}{M} \sum_{i=1}^{M} \mathrm{KL}(P_{\theta_i} \| P_{\theta_0}) \leq \alpha \log M$ with $0 < \alpha \leq 1/8$. Then*

$$\inf_{\hat{F}_n} \sup_{\theta \in \Theta} \mathop{\mathbb{P}}_{D \sim P_\theta} \left\{ |\hat{F}_n(D) - F(\theta)| \geq s \right\} \geq \frac{\sqrt{M}}{1 + \sqrt{M}} \left( 1 - 2\alpha - \sqrt{\frac{2\alpha}{\log M}} \right) > 0.$$

*Proof.* First, it is clear that

$$\inf_{\hat{F}_n} \sup_{\theta \in \Theta} \mathop{\mathbb{P}}_{D \sim P_\theta} \left\{ |\hat{F}_n(D) - F(\theta)| \geq s \right\} \geq \inf_{\hat{F}_n} \max_{\theta \in \{\theta_0, \ldots, \theta_M\}} \mathop{\mathbb{P}}_{D \sim P_\theta} \left\{ |\hat{F}_n(D) - F(\theta)| \geq s \right\}. \quad (14)$$

Next note that if

$$|\hat{F}_n - F(\theta_j)| > \min_{0 \leq k \leq M} |\hat{F}_n - F(\theta_k)|$$

holds for some $j \in \{0, \ldots, M\}$ then using triangle inequality we get

$$2|\hat{F}_n - F(\theta_j)| \geq |\hat{F}_n - F(\theta_j)| + |\hat{F}_n - F(\theta_{i^*})| \geq |F(\theta_{i^*}) - F(\theta_j)| \geq 2s,$$

where $i^* \in \{0, \ldots, M\}$ is defined by

$$|\hat{F}_n - F(\theta_{i^*})| = \min_{0 \leq k \leq M} |\hat{F}_n - F(\theta_k)|.$$

This shows that

$$\mathop{\mathbb{P}}_{D \sim P_{\theta_j}} \left\{ |\hat{F}_n(D) - F(\theta_j)| \geq s \right\} \geq \mathop{\mathbb{P}}_{D \sim P_{\theta_j}} \left\{ |\hat{F}_n(D) - F(\theta_j)| > \min_{0 \leq k \leq M} |\hat{F}_n(D) - F(\theta_k)| \right\}.$$

Together with (14) this inequality leads to

$$\inf_{\hat{F}_n} \sup_{\theta \in \Theta} \mathop{\mathbb{P}}_{D \sim P_\theta} \left\{ |\hat{F}_n - F(\theta)| \geq s \right\} \geq \inf_{\hat{F}_n} \max_{0 \leq j \leq M} \mathop{\mathbb{P}}_{D \sim P_{\theta_j}} \left\{ |\hat{F}_n - F(\theta_j)| > \min_{0 \leq k \leq M} |\hat{F}_n - F(\theta_k)| \right\}.$$

Note that

$$\mathop{\mathbb{P}}_{D \sim P_{\theta_j}} \left\{ |\hat{F}_n(D) - F(\theta_j)| > \min_{0 \leq k \leq M} |\hat{F}_n(D) - F(\theta_k)| \right\}$$

is the probability of mistake for a *minimum distance test* in the problem of testing $M + 1$ hypotheses. Taking infimum over all tests $\Psi \colon \mathcal{X} \to \{0, \ldots, M\}$ we finally obtain

$$\inf_{\hat{F}_n} \sup_{\theta \in \Theta} \mathop{\mathbb{P}}_{D \sim P_\theta} \left\{ |\hat{F}_n(D) - F(\theta)| \geq s \right\} \geq \inf_{\Psi} \max_{0 \leq j \leq M} \mathop{\mathbb{P}}_{D \sim P_{\theta_j}} \{ \Psi \neq j \}.$$

We complete the proof by repeating the remaining steps of the proof of Theorem 2.5 of [22].  □

# B   Auxiliary results

This section contains useful technical results used in the proofs in Section 3.

**Lemma B.1.** *Assume $X$ and $Y$ are independent and both uniformly distributed on $[-a, a]$ for $a > 0$. Then for $t \neq 0$*

$$\mathbb{E}\left[e^{tXY}\right] = \frac{1}{2ta^2}\operatorname{Shi}(ta^2),$$

*where*

$$\operatorname{Shi}(z) = \int_0^z \frac{e^x - e^{-x}}{x}dx$$

*is a hyperbolic sine integral.*

*Proof.* The moment-generating function of a uniform distribution is well known and has the following form:

$$\mathbb{E}\left[e^{tX}\right] = \frac{e^{ta} - e^{-ta}}{2ta} \text{ for } t \neq 0.$$

Using this together with the tower rule of expectation we get

$$\mathbb{E}\left[e^{tXY}\right] = \mathbb{E}_Y \mathbb{E}_X \left[e^{tXY}\big|Y\right] = \mathbb{E}_Y \left[\frac{e^{tYa} - e^{-tYa}}{2tYa}\right]$$

$$= \frac{1}{4a}\int_{-a}^{a} \frac{e^{tya} - e^{-tya}}{tya}dy = \frac{1}{4ta^2}\int_{-ta^2}^{ta^2} \frac{e^y - e^{-y}}{y}dy = \frac{1}{2ta^2}\operatorname{Shi}(ta^2).$$

$\square$

The following result can be found in [2, Exercise 2.3].

**Theorem B.2** (Chebyshev-Cantelli's inequality)**.** *Let $t \geq 0$. Then for any random variable $X$ with finite mean $\mathbb{E}[X]$ and variance $\mathbb{V}[X]$,*

$$\mathbb{P}\{X - \mathbb{E}[X] > t\} \leq \frac{\mathbb{V}[X]}{\mathbb{V}[X] + t^2}.$$