[Reviews · NeurIPS 2016]

Reviewer 1

Summary

The paper deals with the estimation of Maximal Mean Discrepancy (MMD) with radial kernels from the minimax point of view. It gives new lower bounds for the minimax probability, which coincide with the known upper bounds, up to multiplicative constants (depending on the properties of the kernel). A first lower bound for Gaussian kernels is straightforwardly obtained from Le Cam's method, following the same arguments as Tolstikhin, Sriperumbudur and Muandet (2016), but this lower bound depend on the dimension of the data. A second lower bound is developed, not depending on this dimensionality parameter and generalized to radial kernels, based on classical arguments from the minimax theory for functionals (the two fuzzy hypotheses method, as named by Tsybakov (2008)). This new lower bound enables to establish a classical minimax parametric rate of estimation of the MMD, and to prove that the empirical estimator and its $U$-statistic variant, used in the MMD two-sample tests for instance, are optimal in the presented minimax sense.

Qualitative Assessment

The paper is very interesting as it provides the first known lower bounds for the minimax rate of estimation of the MMD distance between two distributions based on finite random samples, considering general radial kernels, which match with the known upper bounds up to constants. Theses bounds lead to a classical parametric rate of estimation, not depending on the dimension of the data. The text could however be more fluent, and easier to read for nonspecialists of the minimax estimation theory. For instance, the results are all expressed with respect to the minimax probabilities. Could the link with the more classical minimax risk be clearly written ? Moreover, to my mind the only main drawback of this work is that it does not tackle the real target issue of minimax testing, which is usually closely related to the minimax estimation of functionals. The obtained parametric rate of estimation is indeed known to coincide with the parametric rates of testing when one considers weak distances (such as the Bounded Lipschitz or the MMD distances) between distributions. Finally, I noticed a few mistakes in the text (in (1) a absolute value bar is lacking, and $r_{n,m,k}(\mathcal{P})$ should be replaced by $r_{n,m,k}(\mathcal{P})^{-1}$ in (2).

Confidence in this Review

2-Confident (read it all; understood it all reasonably well)


Reviewer 2

Summary

The authors provide some minimax lower bounds for the estimation of the maximum mean discrepancy (MMD) based on finite samples in the case of radial kernels on $\mathds R^d$. They first provide a minimax lower bound that depends on the dimension $d$ (Theorem 1), whose proof is based on the Le Cam's method, and hence they get rid of such dependence by using the {\it method of two fuzzy hypothesis} (Theorem 2).

Qualitative Assessment

The paper is well structured: the problem is clearly and exhaustively explained and the techniques used in the paper are well described. The proofs of the two main results are provided Additional comments: > lines 37 - 38: notation $O_p$ but $p$ is not defined > in eq. (1) a $\big|$ is missing > line 178 it is explained that $MMD_k(P_0,Q_0)=0$ but in the previous page in eq. (6) it is not.

Confidence in this Review

2-Confident (read it all; understood it all reasonably well)


Reviewer 3

Summary

This paper processes an estimation problem of the maximal mean discrepancy (MMD), which is a distance between two probability measures given as the RKHS norm of the difference of their kernel mean embeddings. In particular, this paper provides the minimax lower bound over the distributions with C^\infty-densities, for the Gaussian kernel (Theorem 1) and for general radial kernels (Theorem 2). The proofs are technically sound. The authors use a reduction scheme to testing (i.e. the Le Cam method), which is a standard technique for deriving the lower bounds in nonparametric problems.

Qualitative Assessment

Overall, the paper is well written and organized. However, I feel its contribution to the literature is relatively incremental, because of the following two points: (1) Although the minimax rate n^{-1/2} in finite dimensional problems, as mentioned in the paper, cannot be extended straightforwardly to infinite dimensional problems, the rate max( n^{-1/2}, m^{-1/2}) itself is not a surprising result. Because, in general, infinite dimensional estimators have slower rates than n^{-1/2}, and provided the fact that the canonical plug-in estimator MMD_{n, m} attains the rate, the minimax rate optimality of this estimator seems to be quite natural. (For the estimation problem of kernel mean embeddings, the minimax rate of n^{-1/2} seems to have been already proved in Tolstikhin, et al. (2016).) (2) The “statistical” impact of the results is not sufficiently explained in the paper. In practice, the MMD estimation might appear as partial problem of the other statistical applications, such as two-sample testings. I wonder if the lower bounds provided in this paper can be interpreted as unimprovability of some statistical inference problems.

Confidence in this Review

2-Confident (read it all; understood it all reasonably well)


Reviewer 4

Summary

This paper provides minimax lower bounds for MMD estimators. The paper gives two main results, the first one shows that the lower bound scales as O(d^{-1/2}\max(n^{-1/2}, m^{-1/2})). This has a dependence on d, which is eliminated in the second part. The second part gives a lower bound that scales as O(\max(n^{-1/2}, m^{-1/2})), where the constant only depends on the kernel selected. These results show that the empirical and shrinkage estimators are minimax optimal up to the constants

Qualitative Assessment

This is a well written paper with a strong theoretical result, that O(n^{-1/2}+m^{-1/2}) is the best minimax asymptote for MMD estimation. Theorem 2 is particularly impressive as the authors prove that the constant for the asymptote only depends on properties of the kernel, rather than dimensionality of distribution. The authors could try to make the theorems easier to understand. For example, the meaning of $P^n \times Q^n {...}$ takes a while to figure out. Some notation could be very confusing for readers unfamiliar to the area. Despite of the page limits, it could be meaningful to add additional explanation, or even illustrations to make the results easier to read and understand.

Confidence in this Review

2-Confident (read it all; understood it all reasonably well)


Reviewer 5

Summary

This paper presents for the first time lower bounds for the estimation of the Maximum Mean Discrepancy MMD (a distance on the space of probability measures) based on kernel mean elements associated to a measure, given a kernel defining an RKHS. Replacing these by empirical kernel means serves as an empirical estimator for MMD. A first result establishes the minimax rate optimality of the empirical MMD stated for Gaussian kernels in dimension d (but valid in a more general context of radial universal kernels, as the authors remark). However, the lower bound depends on d, making the result weak in high dimension. The main result of the paper is a refined analysis leading to a minimax lower bound, only depending on kernel properties (and thus is independent of d), valid also in the more general case of radial universal kernels. However, both results are stated only for the subclass of probability measures possessing a smooth density. The technically involved proofs follow standard ideas (Le Cam’s method and the method of two fuzzy hypotheses, resp.) relegating some material to the appendix.

Qualitative Assessment

The paper contains interesting new results. It is written very well and all material is organized quite efficiently. In view of the fact that the lower bound in Theorem 2 only depends on the properties of the kernel one might ask why both theorems are restricted to the subclass of probability measures with smooth densities. From the proofs it is not apparent why this should be necessary. Clarification of this minor point could still improve this good paper. Typo: Eq. (1)… a “|” is missing on the left

Confidence in this Review

2-Confident (read it all; understood it all reasonably well)